



# On the Implementation of External Forcings in a Regional Climate Model - A Sensitivity Study around the Samalas Volcanic Eruption in the Eastern Mediterranean / Middle East

Eva Hartmann[1], Mingyue Zhang[1,a], Sebastian Wagner[2], Muralidhar Adakudlu[3,b], Jürg Luterbacher[1,3], and Elena Xoplaki[1,3]

[1]Department of Geography, Climatology, Climate Dynamics and Climate Change, Justus-Liebig University Giessen, Giessen, Germany
[2]Helmholtz-Zentrum Hereon, Geesthacht, Germany
[3]Center for International Development and Environmental Research, Justus-Liebig University Giessen, Giessen, Germany
[a]Now at Max Plank Institute of Meteorology, Hamburg, Germany
[b]Now at Norwegian Meteorological Institute, 0313 Blindern, Oslo, Norway

**Correspondence:** Eva Hartmann (Eva.Hartmann@geogr.uni-giessen.de)

**Abstract.** Natural and human-historical archives document regional climate variations and extremes of the past 2500 years across the Eastern Mediterranean / Middle East. Earth System Models (ESM) can contribute to the interpretation of the variations in the paleoclimate data and the dynamics of the underlying mechanisms. State-of-the-art ESMs have a good temporal resolution but are spatially too coarse to adequately address regional processes. Here we provide for the first time a regional

climate model (RCM) output adjusted to past climates forcings covering the Eastern Mediterranean / Middle East at a $0.44°$ horizontal and up to hourly temporal resolution. CMIP6 external climate forcings of volcanic, orbital, solar and greenhouse-gas changes are implemented in the RCM COSMO-CLM (CCLM, COSMO 5.0 clm16). The sensitivity of the model to each of the implemented forcing is tested separately and in combination in a case study around the large Samalas volcanic eruption (1255-1264 CE) with strong socio-economic impacts in the study area. We evaluate the impact of the different implemented

forcings compared with the standard CCLM model version for the present time. The orbital forcing is found to have the largest effect with cooler winter/spring and warmer autumn during the test period. The volcanic forcing has a strong cooling effect for a couple of years after the large volcanic eruption. Other climate forcings only show a smaller impact in the sensitivity study, while the improvements in simulated precipitation are mainly due to the higher spatial resolution than to a specific forcing. The study is part of the new 2500-year-long transient, fully forced RCM simulation over the Eastern Mediterranean / Middle

East. This work introduces a unique source of information for the comparison of paleoclimate simulations with proxy records and reconstructions. We aim to enhance our understanding of the role of single and joint forcings on climate variability and extremes, their underlying processes at the regional scale, potential climate-society interactions and address limitations and uncertainties.



# 1   Introduction

Earth System Models (ESMs) are powerful tools that combine the complex interactions of physics, chemistry and biology of
Earth systems and contribute to a better understanding of processes in and between its spheres. Various ESMs with a standard
resolution of 1.9° (approximately 200 km in the atmospheric component) are used to investigate specific periods of the past,
present, and future (Giorgetta et al., 2013). The Coupled Model Intercomparison Project Phase 6 (CMIP6) is a collaborative
initiative within the climate science community aimed at advancing our understanding of the Earth's climate system (Eyring

et al., 2016). Systematic experiments conducted within CMIP6 include the study of past climate conditions, projections of
future climate and climate sensitivity, refinement of model performance, validation of models and provision of data to support
decision-making. The Paleoclimate Modeling Intercomparison Project (PMIP) was established with the aim of investigating
the influence of forcing and feedback on the climate system and comparing climate model simulations with paleoclimate re-
constructions and observations (Williams et al., 2020, 2021; Lunt et al., 2021) and is under CMIP6 in its fourth phase (PMIP4)

(Kageyama et al., 2018).

The CMIP6/PMIP4 simulations have a large grid size of 200 x 200 km, thus $400\,\mathrm{km}^2$ with the exact same climate conditions.
This poses considerable challenges for research on the regional climate in areas of complex topography, such as mountainous
or coastal regions. Additionally, it complicates comparison with proxy data, which often has limited spatial coverage and rep-
resentation. Moreover, integrating these outputs into impact studies becomes even more challenging (e.g. Phipps et al. (2013);

Xoplaki et al. (2018)). To address the limitations of coarse ESM simulations for regional climate analysis, researchers apply sta-
tistical and/or dynamical downscaling techniques. Dynamical downscaling employs high-resolution models to capture regional
features, while statistical downscaling leverages relationships between large-scale variables to enhance spatial resolution. The
PALEOLINK working group of the PAGES (Past Global Changes; www.pastglobalchanges.org) network has identified both
approaches as scientific goals (Gómez-Navarro et al., 2019). Ludwig et al. (2019); Gómez-Navarro et al. (2015a, b); Gómez-

Navarro et al. (2019) demonstrate that regional climate simulations obtained through dynamical downscaling can enhance
the comparability between climate model output and paleo climatic evidence at the regional scale. Furthermore, Armstrong
et al. (2019) found improved climatology of regional climate simulations compared to global climate simulations in the North-
ern Hemisphere for the millennium preceding the industrial era (*past1000*), the mid-Holocene around 6,000 years ago (*mid-
Holocene*) and the Last Glacial Maximum around 21,000 years ago (*lgm*). High-resolution RCM simulations offer detailed

insights into general and regional atmospheric circulation patterns (Cortina-Guerra et al., 2021). Such simulations have been
used to investigate the interaction between different regional climates and other spheres (e.g. Ludwig et al. (2017, 2018, 2021);
Ludwig and Hochman (2022); Schaffernicht et al. (2020)) during the *lgm* with the Weather Research and Forecasting Model
(WRF). Additionally, RCMs assist in interpreting proxy data, as illustrated by Pinto and Ludwig (2020). RCM simulations also
facilitate consistency analysis of gridded reconstructions, demonstrated by Gómez-Navarro et al. (2015a, b).

The low-resolution ESM used to drive the RCM, significantly impacts the high-resolution simulations. To mitigate this influ-
ence, external climate forcings from the ESM can be implemented directly in the RCM. This approach enables a more direct
effect of the forcings on the RCM simulation, reducing dependence on the ESM and enhancing understanding of their effects on



the regional climate (Prömmel et al., 2013). Various external climate forcings become relevant, depending on the period under consideration. The forcings that are important over the last 2500 years are the same as in the CMIP6/PMIP4 *past1000* experiments, which are the orbital, solar, volcanic, greenhouse gas and land-use changes (Jungclaus et al., 2017). Those forcings have not yet been explicitly implemented in standard regional climate models. Here, we implemented them in the COSMO-CLM model (Rockel et al., 2008). This method leads to the creation of a novel, detailed paleo-regional climate model (paleo-RCM). To investigate the effects of various climate forcings, a set of decade-long sensitivity experiments is conducted. Simulations are performed with individual forcings applied separately, as well as with combined forcings, and compared to a reference simulation calibrated to present-day conditions. This approach enables the evaluation of the relative contributions of external forcings and internal variability to observed changes (Otto-Bliesner et al., 2016). The RCM is driven by the fully forced PMIP-conform MPI-ESM-LR simulation to isolate the additional effect of implementing the forcings into the RCM. This study aims to assess whether incorporating external forcings into the RCM enhances the realism of regional-scale simulation output. To achieve this, differences between sensitivity experiments (using single and combined forcings), the MPI-ESM-LR, and the reference CCLM simulation are analyzed.

The structure of this study is organized as follows: Section 2, Materials and Methods, details the models, their configurations, and the implemented forcings. The described methods, in general, apply to the entire 2500-year period but are specifically tailored for our target period, the Samalas volcanic eruption selected as a sensitivity study. Section 3, Results and Discussion, focuses on the sensitivity analysis for the Eastern Mediterranean and Middle East during the Samalas period. It presents and interprets simulated temperature and precipitation data, including comparisons with the ESM, along with annual, seasonal, and monthly distributions. Finally, Section 4 provides the Conclusions and outlines potential directions for future research.

## 2 Material and Methods

### 2.1 Period and Domain

A prominent candidate for setting up sensitivity experiments during the last millennia relates to volcanic outbreaks. Specifically large volcanic tropical eruptions exert impacts on the global and regional climate. In this context, the 12th and 13th centuries are notably active periods (Guillet et al. (2023) and reference therein). The eruption of the Samalas volcano on the Indonesian island of Lombok in 1257 stands out as the 5th largest volcanic eruption of the last millennium in terms of sulfate deposition (Lavigne et al., 2013; Guillet et al., 2017, 2023) and in terms of the emissions stands as the greatest volcanogenic gas injection of the Common Era (Vidal et al., 2016). With a Volcanic Explosivity Index of 7, it ranks as one of the most significant eruptions in history (Whelley et al., 2015). The eruption created a 6-7 km wide caldera, known as Segara Anak, replacing the former Samalas mountain (Rachmat et al., 2016). Volcanic aerosols in the stratosphere, measured by the aerosol optical depth (AOD), sharply increased after the eruption and remained elevated for several years, as illustrated in Figure 1, a reconstruction of Toohey and Sigl (2017) for a wavelength of 550 nm (visible light). The substantial amounts of volcanic sulfate aerosols led to stratospheric warming and surface cooling (Robock, 2000; Crowley et al., 2008). Consequently, the Northern Hemisphere experienced some of the coldest summers of the past millennium in the years 1258 and 1259 (Guillet





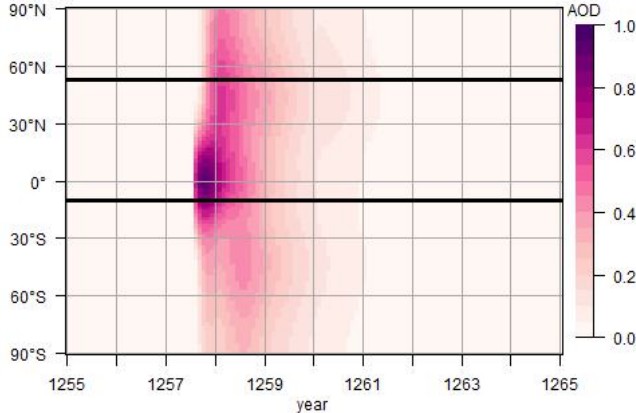

**Figure 1.** AOD from 1255 to 1265 for the different latitudinal bands. The latitudes used for the simulations are between the bars. Own representation of Toohey and Sigl (2017).

et al., 2017). These climatic shifts combined with societal vulnerabilities have been linked to historical instances of agricultural deficits, civil unrest, pestilence, and migration (e.g. Post (1977); Oppenheimer (2011); Luterbacher and Pfister (2015); Guillet et al. (2017, 2023); Malawani et al. (2022)). The effects of the Samalas outbreak were felt throughout the Mediterranean and contributed to considerable cooling and existential crises in various regions (e.g. Xoplaki et al. (2016, 2018, 2021); Guillet

et al. (2020, 2023); Malawani et al. (2022)). The cumulated impact of these events has been suggested as a contributing factor to the onset of the Little Ice Age (Miller et al., 2012). In our study, we focus on the decade spanning 1255-1264 CE which includes the Samalas eruption in 1257, as a test case for conducting sensitivity experiments.

The simulations cover a domain similar to MENA-CORDEX (Zittis et al., 2014), with a focus on the Eastern Mediterranean and the Middle East (EMME), which includes the Eastern Mediterranean, North-East Africa, the Arabian Peninsula, and the

Middle East - a region rich in historical significance. Cultures and societies that flourished in this area, such as Ancient Egypt, Ancient Greece, and Ancient Rome, had a profound influence worldwide (Wilson, 2013; Sallares et al., 1991; Feldman, 1991). These societies expanded their reach beyond Europe, extending along the Nile River, the Red Sea, and toward the Caspian Sea and the Persian Gulf. Given the extensive history of developed cultures, abundant historical data and proxy records are available in this region. The EMME is of interest not only in terms of history and society but also in terms of climate, which

may be closely linked to historical and societal developments and changes (Crowley and North, 1988; Holmgren et al., 2016; Izdebski et al., 2016). In addition, the area is recognized as a climate change hotspot (Giorgi, 2006; Lelieveld et al., 2012, 2016; Cramer et al., 2018; Zittis and Hadjinicolaou, 2017; Zittis et al., 2019, 2022), making it particularly compelling for climatic studies.



## 2.2 MPI-ESM-LR

We performed a global simulation using the MPI-ESM-LR (Giorgetta et al., 2013), which is a new realization of the transient MPI-ESM simulation following the CMIP6-protocol for PMIP4 past1000 simulations by Jungclaus et al. (2017). To suit the requirements of the regional climate model COSMO-CLM, the output intervals are adjusted accordingly with variables written out at a 6-hourly resolution. The MPI-ESM-LR comprises the coupled general circulation models for the atmosphere - ECHAM6 (Stevens et al., 2013) and the ocean - MPIOM (Jungclaus et al., 2013), along with the subsystem models for land and

vegetation JSBACH (Reick et al., 2013; Schneck et al., 2013) and marine biogeochemistry HAMOCC5 (Ilyina et al., 2013). Atmospheric and vegetation-related variables in particular are essential inputs for the regional climate simulation. The spatial resolution of the simulation is $1.875°$ which is approximately equivalent to 200 km (T63).

## 2.3 COSMO-CLM

The regional climate simulations are performed with the regional climate model COSMO-CLM (Rockel et al., 2008), which is

the model developed by the COnsortium for Small-scale MOdelling (COSMO) in CLimate Mode (Baldauf et al., 2011; Rockel et al., 2008) and is further developed by the CLM-Community. In this study, the COSMO model version 5.0 with CLM version 16 (COSMO-CLM $- v5.0\_clm16$) is used. The model is forced by a transient MPI-ESM-LR simulation (Jungclaus et al., 2017) and the interpolation from the forcing data to the model is performed using INT2LM version 2.05 with CLM version 1 (INT2LM $- v2.05\_clm1$) (Schättler and Blahak, 2017). Time integration is achieved using the two time-level Runge-Kutta

scheme (Jameson et al., 1981) with a model time step of 300 seconds. Convection parameterization is based on the Tiedtke scheme (Tiedtke, 1988) is used. The representation of albedo and aerosols, identified as crucial parameters by Bucchignani et al. (2016), are set according to their values. The land surface model is TERRA-ML (Doms et al., 2011; Schulz et al., 2016). The external data set is prepared using EXTPAR (Smiatek et al., 2008). These settings remain consistent for all experiments, facilitating a pure sensitivity study solely focused on differences attributable to external forcings.

The simulations are carried out for a domain including the Eastern Mediterranean, the Middle East and the Nile River basin from Lake Victoria to the Delta ($Lon = 4° - 60°E$, $Lat = 5°S - 49°N$). In this study only the EMME region is analyzed ($Lon = 18° - 60°E$, $Lat = 12.5° - 42.5°N$). The simulation and the analyzed domains are shown in Figure 2 with a mesh size of $0.44°$ ($\sim 50\,km$). Each experiment spans the period from 1255 to 1264 CE.

## 2.4 External Climate Forcings

The external forcings are defined based on the recommendations for the PMIP4 past1000 contribution to CMIP6 (Jungclaus et al., 2017). These forcings include time-varying parameters. Orbital forcing is crucial for long time scales spanning centuries to millennia (Rial and Anaclerio, 2000; Cubasch et al., 2006). Solar, greenhouse gas and land-use change forcings exhibit effects within decades, albeit indirectly, leading to a comparably slower climate response (Cubasch et al., 2006). In contrast, volcanic forcing has a strong, rapid and short-term impact (Wigley et al., 2005). The volcanic forcing is described by varia-

tions in aerosol optical depth (AOD) at a wavelength of 550nm by Toohey and Sigl (2017). AOD is the only climatic forcing





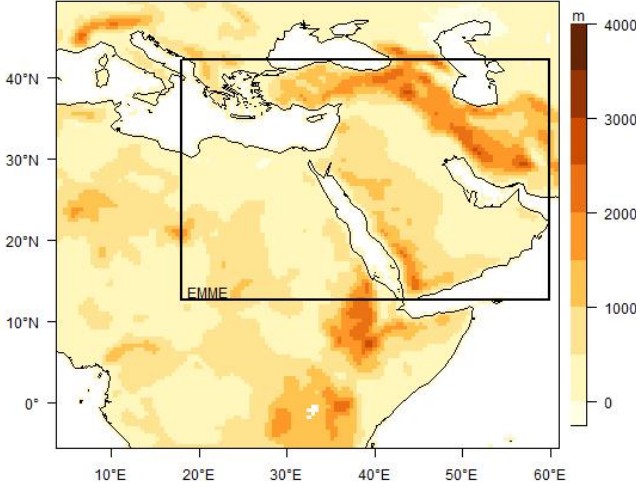

**Figure 2.** Simulation domain with topography and the EMME region in the black rectangle.

significantly varying during the simulated period, with the eruption in 1257 registering as a prominent peak of AOD, one of the largest in the Common Era (Figure 3). The solar forcing is represented by changes in the total solar irradiance (TSI), which is influenced by sunspots and faculae directly on the sun's surface by the SATIRE reconstruction data set (Jungclaus et al., 2017). The present-day TSI values closely match those used in the Samalas case study and exhibit minimal variability over the ten

years, as depicted in Figure 3. In contrast, the TSI value used for the reference simulation differs significantly from the Samalas period values due to the specifications of the CCLM model for present-day conditions. Orbital forcing is represented by the eccentricity, the obliquity and the longitude of perihelion, varying on time scales of dozens of millennia. A static implementation of orbital forcing initially prepared by Patrick Ludwig (pers. comm.) is extended throughout this study. Annually data is provided by Berger (1978). Compared to the mid-13th century, today's orbit exhibits slightly lower eccentricity and obliquity,

and higher precession. Figure 4 illustrates the cumulative effect of these factors on insolation during the 13th century (1250 CE) compared to the present (1950 CE, reference configuration). In the Eastern Mediterranean and Middle East (EMME) region, this translates to lower insolation from December to June during the 1260s, while July to November experienced higher insolation compared to the present. GHG concentrations encompassing $CO_2$, $CH_4$ and $N_2O$ are implemented via equivalent $CO_2$, all obtained from (Meinshausen et al., 2017). It is noteworthy that the $CO_2$ concentration in the reference simulation (330 ppm)

is lower than the combined effective $CO_2$ utilized in the transient GHG forcing as illustrated in Figure 3. In this study only results for the sensitivity period (1255 to 1264 CE) are presented, but Figure 3 also shows the evolution of the external climate forcings over the last 2.5 millennia to illustrate and contrast the long- and short-term changes in individual forcings.

The choice of land-use data source significantly influences atmospheric conditions (Zhang et al., 2021). The CCLM cannot differentiate between various land cover types except for deciduous and evergreen forests and tends to overestimate the effect

of the shrubs and grass due to every plant following the same phenological cycle (Hartmann et al., 2020). To address this limitation, a transient land-use dataset based on global JSBACH output was implemented in the CCLM. Although not utilized in





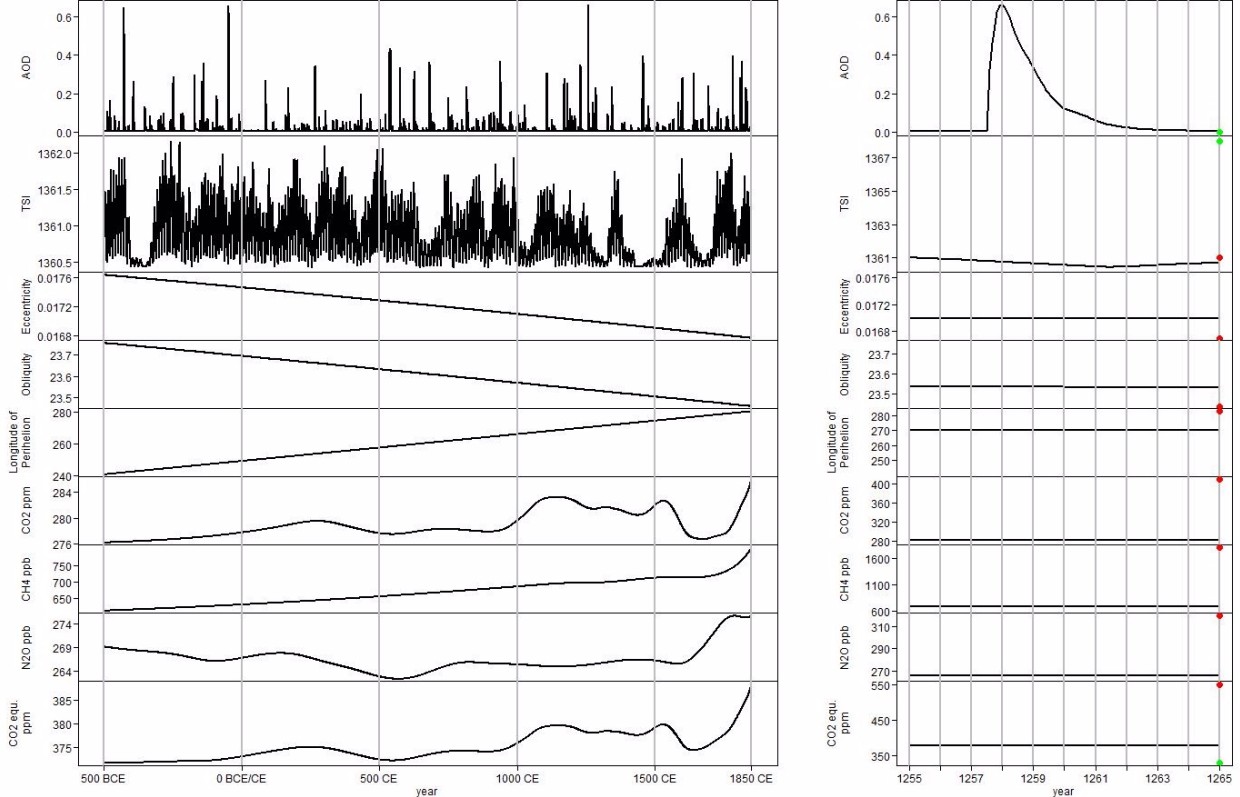

**Figure 3.** Transient Forcings for two distinct periods: 500 BCE to 1850 CE (left) and 1255-1265 CE (right). Present-day value (2020 CE) is denoted by a red dot, and the value utilized in the reference COSMO-CLM simulation by a green dot (AOD, TSI, $CO_2$-equivalent). The components of the forcings are as follows: Aerosol Optical Depth (AOD) by Toohey and Sigl (2017) representing volcanic forcing, Total Solar Irradiance (TSI) by Jungclaus et al. (2017) representing solar forcing, Eccentricity, Obliquity and Longitude of Perihelion by Berger (1978) representing orbital forcing and Effective $CO_2$ ($CO_2$, $CH_4$ and $N_2O$) representing greenhouse gas forcing (Meinshausen et al., 2017).

this study to avoid confounding the influence of other forcings, details about the land-use forcing can be found in Appendix A since a transient land-use forcing is used in our simulation of the last 2500 years. Other potential forcings such as ice sheets or tropospheric aerosols are not specifically implemented from external datasets in either the CCLM or MPI-ESM-LR. External

forcings used in CMIP6 for other historical periods could similarly be incorporated into the CCLM to enable high-resolution simulations of these periods, depending on the scientific interest.

## 2.5 Experiments

To investigate the influence of different forcings on the climate across the EMME region from 1255 to 1264, we conducted simulations using single forcings and their combination, comparing them with the CCLM standard configuration. The CCLM

standard configuration, tuned for present-day climate serves as the reference simulation (REF). Notably, all experiments use





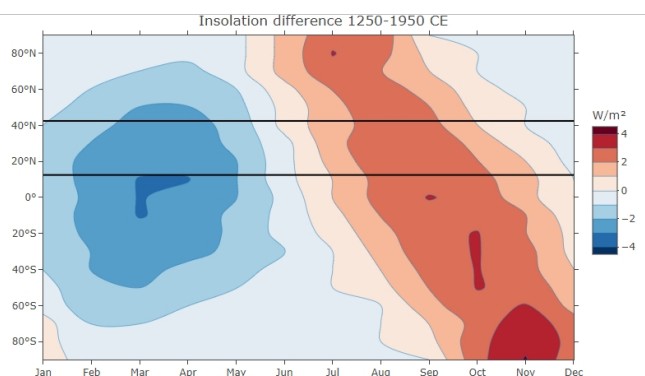

**Figure 4.** Insolation difference due to orbital forcing between the century of the Samalas eruption and present days (1250-1950 CE; own representation of data from Berger and Loutre (1991) linearly interpolated). The study domain is between 12.5 and 42.5 °N marked with horizontal lines.

**Table 1.** Experiments. The MPI-ESM-LR is the driving model, 'REF' is the CCLM simulation with standard configuration, 'ORB/SOL/VOL/GHG' are the simulations with the respective forcing only and 'FULL' is the simulation with all forcings combined. Values for 'transient' can be seen in Figure 3. 'solc' is short for solar constant. Transient values are during the ten-year study period approximately: $CO_2$-equivalent = 378 ppm, solc = 1361 W/m$^2$, AOD-max = 0.6638, eccentricity = 0.017, obliquity = 23.54, perihelion = 270. Present-day orbital values are eccentricity = 0.0167, obliquity = 23.44, perihelion = 282.

| Name | Resolution | Land-use | GHG | Orbital | Solar | Volcanic |
|---|---|---|---|---|---|---|
| MPI-ESM-LR | 1.875° | transient | transient | transient | transient | transient |
| REF | 0.44° | EXTPAR | CO2 = 330 ppm | present-day | solc = 1368 W/m$^2$ | AOD = 0.045 |
| ORB | 0.44° | EXTPAR | CO2 = 330 ppm | transient | solc = 1368 W/m$^2$ | AOD = 0.045 |
| SOL | 0.44° | EXTPAR | CO2 = 330 ppm | present-day | transient | AOD = 0.045 |
| VOL | 0.44° | EXTPAR | CO2 = 330 ppm | present-day | solc = 1368 W/m$^2$ | transient |
| GHG | 0.44° | EXTPAR | transient | present-day | solc = 1368 W/m$^2$ | AOD = 0.045 |
| FULL | 0.44° | EXTPAR | transient | transient | transient | transient |



the same boundary forcing of the fully forced ESM simulation, precluding individual forced sensitivity studies for the ESM. All RCM simulations employ EXTPAR external land-use data and share the same horizontal and temporal resolution and settings, except for the forcing data. Our reference simulation maintains fixed values for GHG concentration (330 ppm), TSI (1368 W/m$^2$) and stratospheric AOD (0.045). The orbital forcing is not explicitly addressed by the model and is designed

for present-day simulations (eccentricity = 0.0167, obliquity = 23.44, perihelion = 282). Explicit values of the forcings are summarized in Table 1, while details of the transient forcings are depicted in Figure 3. For the sensitivity simulations, the orbital, solar, volcanic and GHG forcings are implemented in a transient mode using the introduced datasets. Each forcing is individually incorporated into the CCLM source code, enabling direct assessment of their influence on the simulated climate. For the fully forced simulation, all transient forcings are combined, ensuring consistency between the global ESM and the

high-resolution RCM. This approach aims to yield the most realistic results.

## 3    Results and Discussion

### 3.1    2m Temperature

#### 3.1.1    Comparison ESM - RCM

The higher resolution of CCLM, compared to the MPI-ESM-LR, allows for a more detailed depiction of topographic features,

including coastlines and complex mountain ranges. Figure 5 illustrates the decadal mean 2m temperature and standard deviation of the MPI-ESM-LR and the CCLM models for the period 1255-1264 CE. While there is a similar overall pattern across all simulations, CCLM captures more detailed features in mountainous regions and along the Mediterranean/Black Sea coast. The temperature distribution is influenced by orography in both model simulations, with higher heterogeneity in CCLM due to its higher resolution. Profound differences are observed in regions such as the Red Sea and Caspian Sea coasts, where the CCLM

simulation shows lower temperature values compared to MPI-ESM-LR. Both models present lower standard deviations over the Arabian Peninsula and the Middle East, while the RCM emphasizes higher variability over land compared to sea regions. In summary, while the overall patterns of 2m temperature in the ESM and RCM simulations are generally consistent, substantial differences arise in areas with complex terrain, which are prevalent throughout the EMME region.

The violin plot on the right side of Figure 5 displays all monthly mean temperatures across the entire EMME domain for the 10

years. Notably, the higher resolution leads to an increased frequency of cold temperatures, which can be attributed to elevated regions where lower temperatures are better captured due to the more detailed representation of orography. This phenomenon is explained by the Boltzmann barometric equation, which describes the relationship between atmospheric pressure and altitude. According to the ideal gas law, higher altitudes are associated with lower atmospheric pressure, leading to lower temperatures, with an environmental lapse rate typically assumed to be -6.5 K/km. Consequently, the finer resolution of CCLM yields lower

temperatures in regions characterized by higher altitudes. The temperature follows a bimodal distribution with the major mode between the median and the upper quartile and the minor mode close to the lower quartile. This is indicated by the two climate classifications mainly represented in this large domain, which are the hot arid desert climate in the Sahara and the





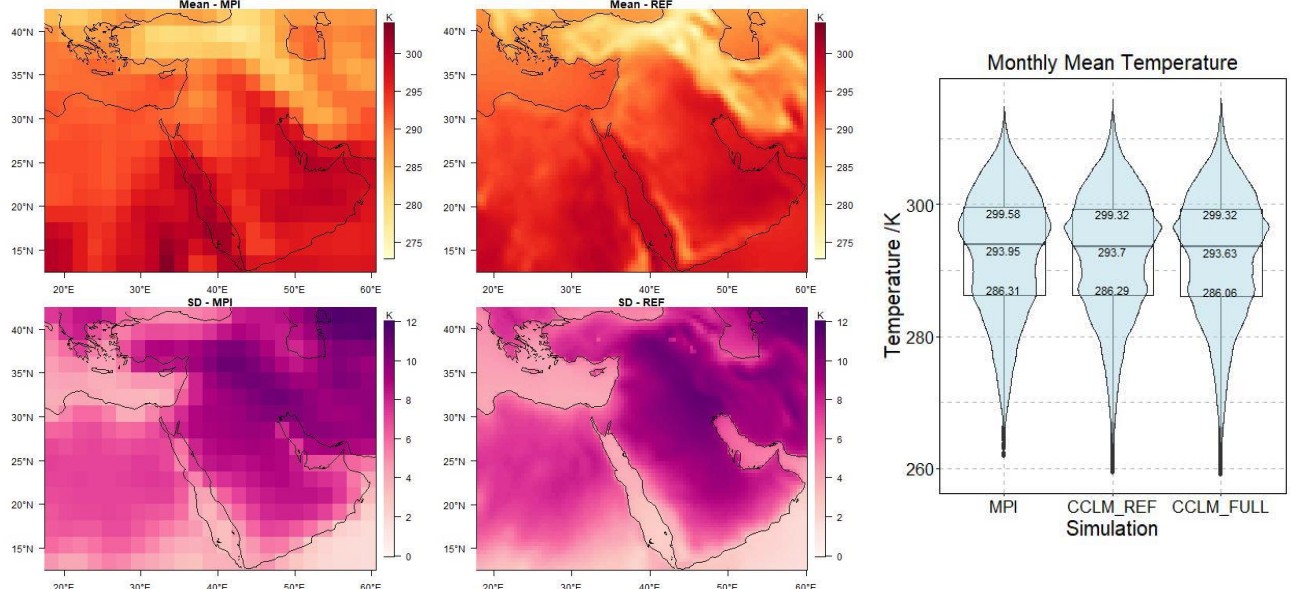

**Figure 5.** Decadal mean (top) and standard deviation (bottom) of 2m temperature in K for the period 1255 to 1264 in the driving MPI-ESM-LR (left) and the fully forced CCLM (center) and violin plot together with boxplot of the monthly mean 2m temperatures across the full EMME domain and decade (right, values are upper quartile, median, lower quartile).

Arabian Peninsula and the warm temperate summer dry climate in the Mediterranean (Kottek et al., 2006). The lower quartile values presented in the box plot and the violin plot on the right side of Figure 5 indicate that the fully forced RCM model

(FULL) exhibits even more extreme temperatures than the standard (REF) configuration. In comparison to the MPI-ESM-LR simulation, the quartile values of the REF and the FULL simulation are between 0.25 K and 0.32 K cooler, indicating a general shift towards cooler temperatures when using the RCM. Further clarifications regarding the differences between the CCLM simulations will be discussed in subsequent sections.

### 3.1.2 Annual and Seasonal Distribution

A major volcanic eruption such as the Samalas in 1257 CE released substantial amounts of sulfate aerosols into the stratosphere, inducing large-scale surface cooling (Robock, 2000). This cooling effect persists as long as the additional volcanic aerosols remain present and physically active in the stratosphere. Figure 6 presents the post-eruption cooling observed in the mean annual temperature over the EMME region. In 1258, the annual mean temperature was more than 1 K cooler than in 1257. By 1259, the concentration of the stratospheric aerosols began to decline again (see also Figure 1), initiating a gradual return to

pre-eruption temperature levels between 1260 and 1263. This observed pattern is in agreement with proxy-based and historical records, as documented by Guillet et al. (2017) and those specific to the EMME region by Xoplaki et al. (2018, 2021).

In the right part of Figure 6, the differences among the sensitivity experiments with respect to the reference simulation (REF) are




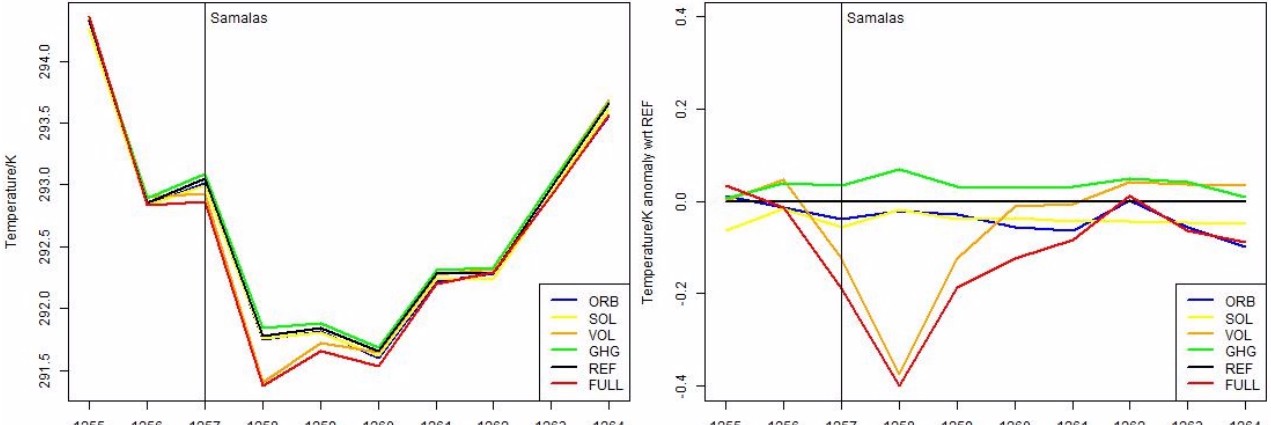

**Figure 6.** Annual mean 2m temperature averaged over the EMME domain (left) for the simulations with the different forcings and the same values represented as anomaly (right) with respect to the reference CCLM from 1255 to 1264.

shown. The VOL and FULL simulations represent the lowest temperatures as they are most influenced by the volcanic eruption with direct implementation of the volcanic forcing. Although all CCLM simulations receive information about the volcanic eruption from the driving ESM at the boundaries, only VOL and FULL explicitly account for the decrease in AOD caused by the volcanic eruption within their radiation routines. In 1258, the first year after the eruption, the annual mean temperature in the FULL and VOL simulations was more than 1.5 K cooler than before the eruption and about 0.4 K cooler than the REF simulation. By 1259, the second year after the eruption, the temperature differences started to decrease, with less than 0.2 K difference, and nearly disappeared in the subsequent years. Across all simulations, the annual mean 2m temperature remains below the pre-eruption value until 1263. This suggests that the direct implementation of the volcanic forcing in the radiation routine of CCLM offers the most significant impact during the initial months to two years following the eruption, aligning with paleo climatic evidence from the region (e.g. Xoplaki et al. (2018, 2021)). The other sensitivity experiments exert a smaller influence on the annual mean temperature compared to the REF simulation. The GHG simulation shows slightly higher temperatures, while the ORB and SOL simulations reveal slightly lower 2m temperatures compared to the REF simulation. The small differences between GHG and SOL compared to REF can be attributed to the slightly higher effective $CO_2$ (330 vs. 378 ppm) and lower TSI (1368 vs. 1361 W/m$^2$) of the implemented forcings compared to the values of the REF simulation, respectively. The difference due to orbital forcing stems from the shift between the 20th-century orbital settings in REF and the 13th-century values in the ORB and FULL simulations. In summary, the FULL simulation reproduces the lowest temperatures during the studied decade as the effects of the different forcings are aggregated. In this context, the positive temperature anomalies due to the GHG forcing are outweighed by the negative temperature anomalies due to solar, orbital, and most importantly, volcanic forcing.

The spatial distribution of annual means (Figure B1), corresponding to the right side of Figure 6, reveals no unexpected or significant variations. Across most of the domain, temperatures are between 0.5 K and 1 K cooler, with the VOL and FULL





simulations for the year 1258, following the Samalas volcanic eruption, showing the most pronounced differences compared to

the REF simulation (as detailed in Figure B1 in Appendix B). Spatial differences between the SOL, ORB, and GHG simulations compared to the REF simulation are minimal, with a few exceptions occurring in specific years, areas, and simulations, showing no consistent pattern.

Figure 7 presents the seasonal mean 2m temperature of the different experiments in comparison to the seasonal mean 2m temperature of REF. In the FULL and ORB simulations, the differences are largest with up to 1.5 K and only here statistically

significant according to a two-sided Student's t-test ($\alpha$=0.05). The orbital forcing induces pronounced negative temperature anomalies during winter and spring over all land areas, attributed to the reduced insolation (see Figure 4). The largest and partly statistically significant differences (more than 1 K cooler) are observed in spring over Mesopotamia for the ORB and FULL simulations. In winter, the temperature differences are smaller, statistically significant only in the FULL simulation, and more pronounced in southern regions such as Arabia and the Sahara. In summer, the ORB simulation, along with the

FULL and GHG simulations to a lesser extent, shows higher temperatures in the northern part of the domain and cooler temperatures in the south. In autumn, temperature anomalies in the ORB and FULL simulations are statistically significant and positive across all land areas, driven by increased insolation (as shown in Figure 4). Deviations exceeding 1 K are most prominent over the Sahara and the Arabian Peninsula. In contrast, the SOL, VOL and GHG simulations show only minor, non-statistically significant temperature anomalies, ranging from slightly negative to positive values below 0.5 K. The FULL

simulation represents a combination of all individually forced simulations, with orbital forcing playing a dominant role in shaping seasonal temperature patterns. The simulations with implemented orbital forcing (ORB, FULL) show cooler winters and springs but warmer summers and autumns over almost the entire domain compared to the REF simulation. Since the REF simulation reflects 1950 CE orbital parameters, comparing simulations with implemented orbital forcing (ORB, FULL) with REF is equivalent to comparing 13th vs. 20th century climate conditions. Over these centuries, the gradual changes induced

by orbital forcing accumulate and can reach several W/m$^2$. Since our domain is primarily situated in the subtropics, which are substantially less affected by atmospheric circulation compared to mid- and high-latitudes, the direct impact of orbital forcing, particularly on 2m temperature, is understandable.

### 3.1.3 Monthly Means - Distribution and Extremes

RCMs have, due to their higher spatial resolution, a higher ability to produce extreme events and values than ESMs (see

Figure 5). To study the influence of different forcings on these extremes, we examine the distributions of monthly mean 2m temperature for the various simulations in each grid box and month within the EMME domain and decade, as illustrated in Figure 8. The shape of the distributions remains consistent across all simulations, indicating similar average and quantile values for temperature. The medians and upper quartiles only slightly vary with values between 293.63 K in FULL to 293.75 K in GHG simulations, respectively, 299.28 K in SOL to 299.4 K in ORB simulations. Although the lower quartiles exhibit a greater

spread, the simulations remain compatible, ranging from 286.06 K in FULL to 286.33 K in GHG.

The density plots in Figure 8 illustrate the distribution of monthly mean temperature across the entire domain in the REF (red) and the FULL (turquoise) CCLM simulations. The density plot of the FULL simulation presents a slight shift towards lower



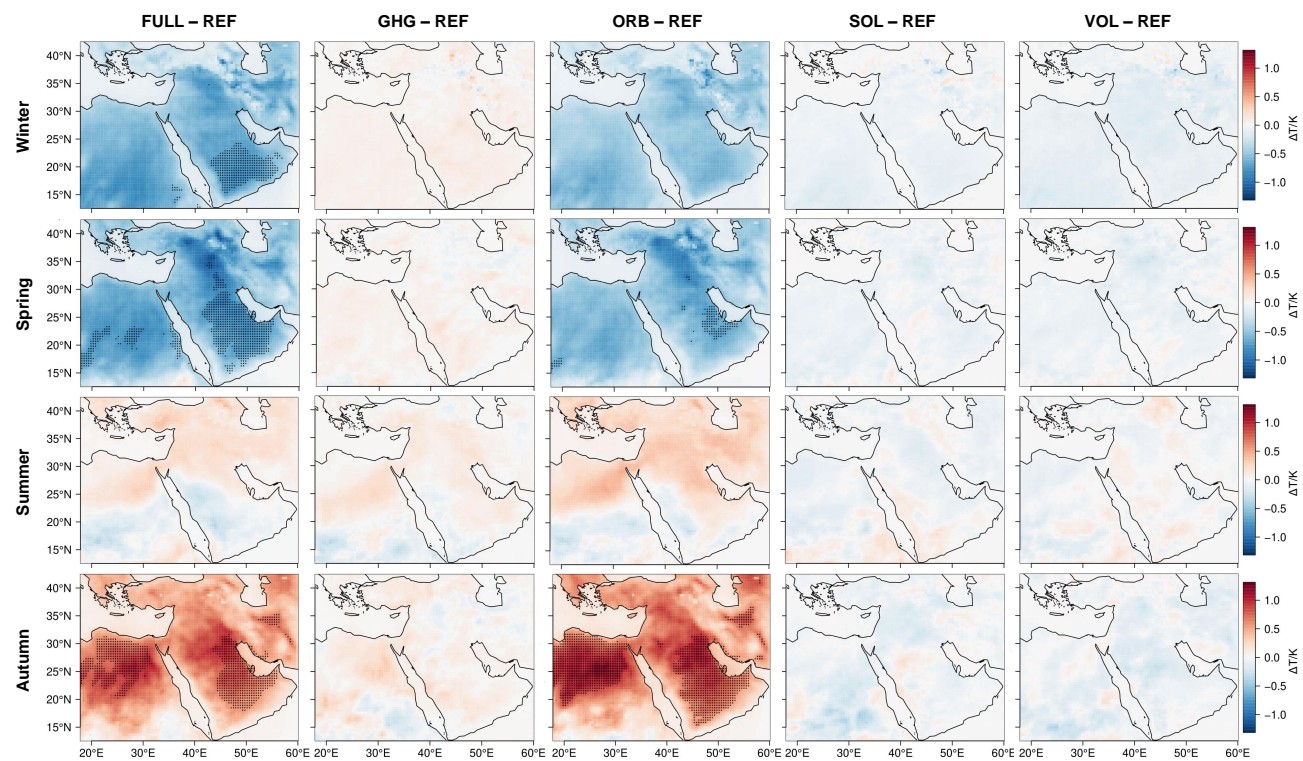

**Figure 7.** Mean seasonal temperature for the decade 1255-1264 for the different experiments compared to REF. Significant differences are marked with a dot (Student's t-test, $\alpha$=0.05). From left to right - FULL, GHG, ORB, SOL, VOL and top to bottom - winter, spring, summer, autumn.

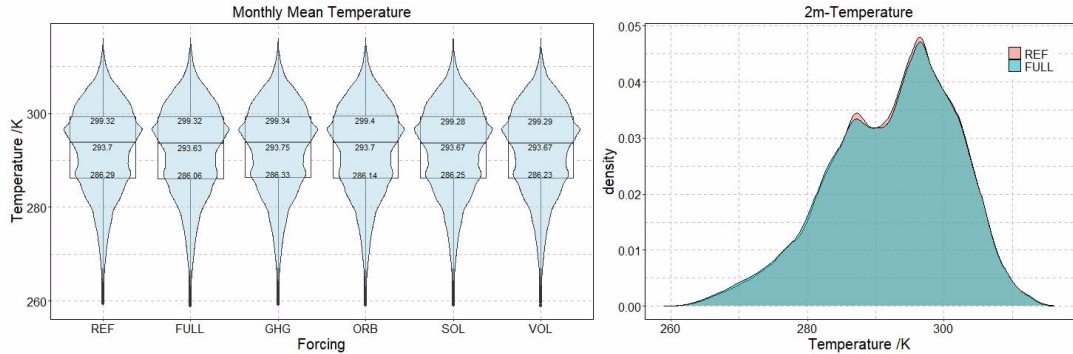

**Figure 8.** Violin plot and boxplot (left, values are upper quartile, median, lower quartile) of the 2m temperature for the different sensitivity experiments and density plot (right) of the 2m temperature for the REF and the FULL CCLM simulations.





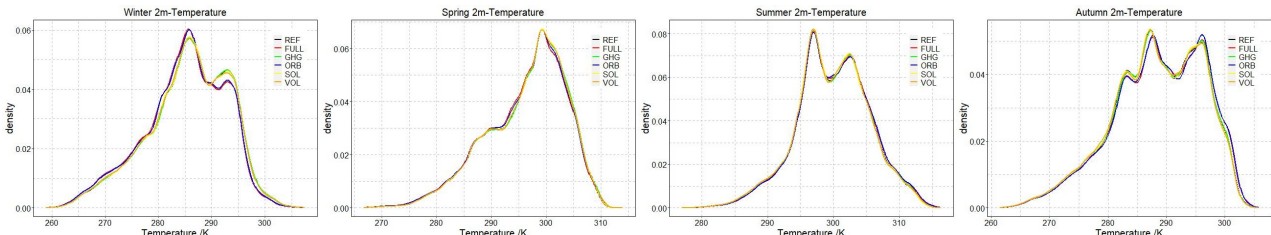

**Figure 9.** Density plot of the 2m temperature for the different forcing experiments in the CCLM for each season.

temperatures, between 260 and 285 K, whereas the REF simulation reveals more moderate temperatures spanning from 285 to 300 K. When the monthly values are separated by seasons, the winter temperature density displays the most pronounced

variations among the simulations (Figure 9). Specifically, the distributions of the FULL and ORB simulations are skewed towards cooler temperatures compared to the other simulations. However, in spring and summer, the simulations demonstrate considerable similarity, and in autumn they closely align, albeit with slight shifts towards warmer temperatures in the FULL and ORB simulations. These results are consistent with the seasonal distribution in Figure 7. An analysis of the spatial distribution of minimum and maximum seasonal 2m temperatures (Figure C1 in Appendix C) supports the previously discussed findings.

The cooling effect of orbital forcing (ORB) is most evident in the minimum temperature distribution over Anatolia during spring, while volcanic forcing (VOL) has the greatest impact on autumn temperatures, particularly in the southern Caucasus region. The strongest effect on maximum temperatures is observed in the ORB simulation and consequently in the FULL simulation. In these simulations, maximum temperatures are lower in winter and spring and higher in summer and autumn. The cooling effect in winter covers the entire domain, whereas the warming observed in autumn does not extend to the Sahara

desert regions.

The analysis of the monthly minimum and maximum values provides insights into temperature extremes. In addition to these extremes, Table 2 shows the 5th, 10th, 90th and 95th percentiles. The lowest temperatures are simulated with the VOL followed by the FULL simulation, which presents the lowest 5th and 10th percentile values, followed by the ORB simulation. On the other hand, the highest temperatures are observed in the ORB simulation, followed by the GHG (10th percentile), and the FULL

simulation (5th percentile and maximum). Overall, the FULL and ORB simulations represent the most extreme temperature values, whereas, the REF, SOL and GHG simulations reflect more moderate values.

## 3.2 Precipitation

### 3.2.1 Comparison ESM - RCM

The general surface cooling caused by volcanic aerosols after the 1257 Samalas eruption is accompanied by various impacts

on atmospheric circulation, including changes in atmospheric humidity and precipitation patterns. The reduction in surface temperatures increases atmospheric stability, leading to decreased convection and evapotranspiration. This, in turn, results in lower atmospheric humidity and reduced precipitation following a major volcanic event on a global scale. On the regional





**Table 2.** Minimum and maximum monthly mean temperature values across the entire domain and the whole decade of the different experiments and 5th, 10th, 90th and 95th percentiles. The experiments with the most extreme values are denoted in bold.

| 2m Temperature/K | | | | | | |
| --- | --- | --- | --- | --- | --- | --- |
| Name | Min | 5% | 10% | 90% | 95% | Max |
| REF | 259.39 | 275.94 | 280.19 | 303.60 | 305.78 | 315.98 |
| FULL | 259.01 | **275.56** | **279.87** | 303.64 | 305.83 | 316.17 |
| GHG | 259.19 | 275.96 | 280.22 | 303.64 | 305.82 | 315.97 |
| ORB | 259.04 | 275.65 | 279.99 | **303.69** | **305.92** | **316.21** |
| SOL | 259.16 | 275.91 | 280.16 | 303.56 | 305.73 | 315.87 |
| VOL | **258.93** | 275.85 | 280.08 | 303.59 | 305.76 | 315.99 |

scale, this pattern might be modified or even reversed because of changes in major regional circulation systems (i.e. monsoon, changes in ITCZ). The higher resolved topography of the CCLM is expected to influence the spatial distribution of precipitation
compared to the ESM. Figure 10 represents the mean annual precipitation sum and standard deviation for both models. While precipitation patterns exhibit similarities, significant deviations are observed in regions with pronounced orographic gradients and/or complex coastlines, owing to the higher resolution of the CCLM. In areas temporally influenced by the ITCZ, the CCLM shows significantly higher precipitation totals and standard deviations compared to the MPI-ESM-LR. Additionally, the CCLM effectively distinguishes between land and sea areas, as evident by higher precipitation totals along the coasts,
especially over the Balkans, the Black Sea, the Caspian Sea and the Red Sea. The effect of higher resolution on precipitation is more complex than for temperature, due to smaller scale thermodynamical, hydrological and cloud microphysical processes, what is partly also connected to higher altitudes. The stark contrast between the sea-level coasts and coastal mountainous regions leads to a forced uplift of air masses, leading to condensation and subsequent precipitation. This effect is particularly notable on the luv sides of the mountains, notably along the Adriatic east coast and the east coast of the Black Sea, with
increased precipitation.

The right part of Figure 10 shows the violin and box plots of the total monthly precipitation. In many regions with no precipitation, the lower quartile of monthly precipitation in both MPI-ESM-LR and CCLM simulations is zero, making the violin and box plots appear less distinct. However, differences become evident in the median and upper quartile values. In the CCLM simulations (1.05 mm/month for REF and 1.02 mm/month for FULL), the median precipitation is approximately
three times higher than that in the MPI-ESM-LR (0.31 mm/month). Similarly, the upper quartile of the precipitation sum in the CCLM (22.07 mm/month for REF and 22.08 mm/month for FULL) is almost double that in the ESM (12.84 mm/month). Furthermore, outliers in the CCLM simulations reach values more than three times higher than those in the ESM simulation.

### 3.2.2 Annual and Seasonal Distribution

Figure 11, left, illustrates the mean monthly precipitation sum for 1255 to 1264. In 1258, the first year after the eruption,
there was a clear decrease in precipitation compared to 1257 (the eruption occurred in September), with a reduction of up to



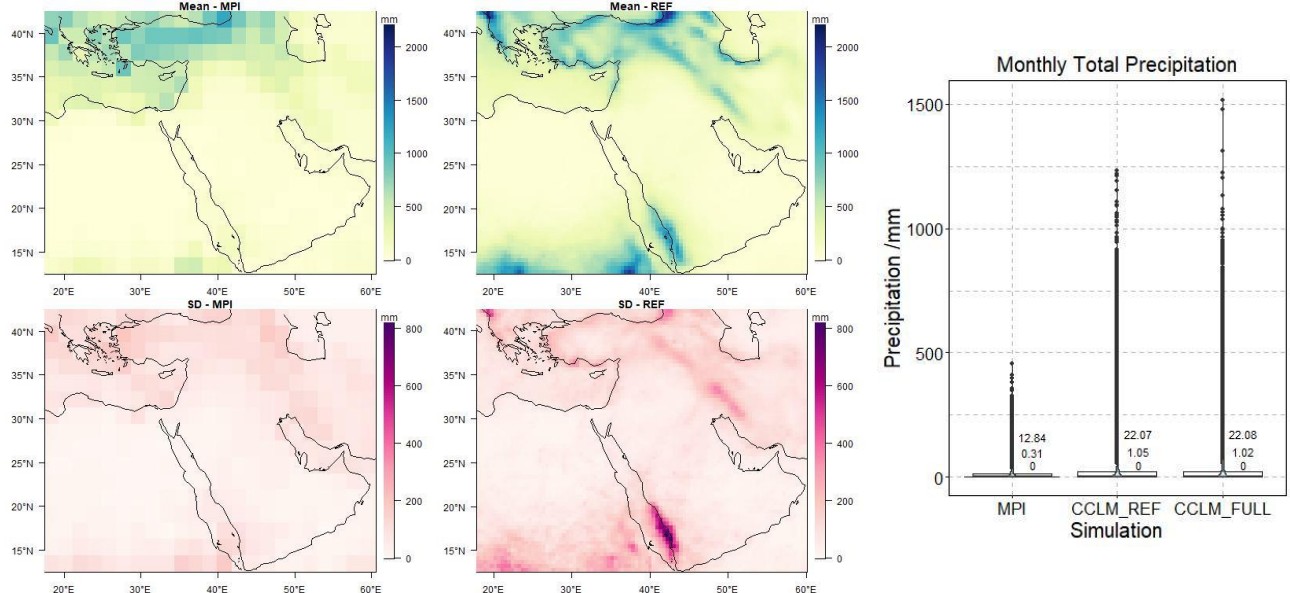

**Figure 10.** Mean (top) and Standard Deviation (bottom) of total annual precipitation in mm for the period 1255 to 1264 in the driving MPI-ESM-LR (left) and the fully forced CCLM (center) and violin plot together with boxplot of the total monthly precipitation sums of the full EMME domain and decade (right, values are upper quartile, median, lower quartile).

3 mm/month (approx. 15 ‰). The reduction is most pronounced in the VOL and FULL simulations. The lowest precipitation amount of the decade is observed in 1255, which predates the eruption. Conversely, the year 1260, the third year after the eruption, exhibits the highest precipitation values of the 10 years.

The differences between the mean monthly precipitation sums of the sensitivity simulations and REF are presented in Fig-
320 ure 11, right. Before the Samalas eruption, the VOL simulation shows similar precipitation amounts to REF. However, after the eruption in 1258, the difference between the simulations reached its peak with an average reduction of about 1 mm/month in the VOL and FULL simulations. The volcanic forcing induces a decrease in precipitation due to surface cooling, resulting in increased atmospheric stability and reduced evapotranspiration. The influence of the volcanic eruption disappears after one to two years. While the FULL simulation reflects the cumulative effect of the individually forced simulations, the ORB simulation
325 demonstrates higher precipitation levels, whereas the GHG and SOL simulations show only minor differences.

The spatial distribution of those mean values for the different years of the decade 1255 to 1264 and sensitivity simulations with respect to REF (Figure B2 in Appendix B) present similar outcomes as Figure 11. The differences are most pronounced in regions influenced by the tropical circulation, where mean precipitation levels are typically highest. Following the volcanic eruption, neither the FULL nor the VOL simulations reveal distinct differences compared to the REF simulation. The precipi-
330 tation differences fall within the range of natural fluctuations and lack the structured temporal and spatial patterns observed in temperature, indicating no clear or strong signal attributed to an individual or combined set of forcings.



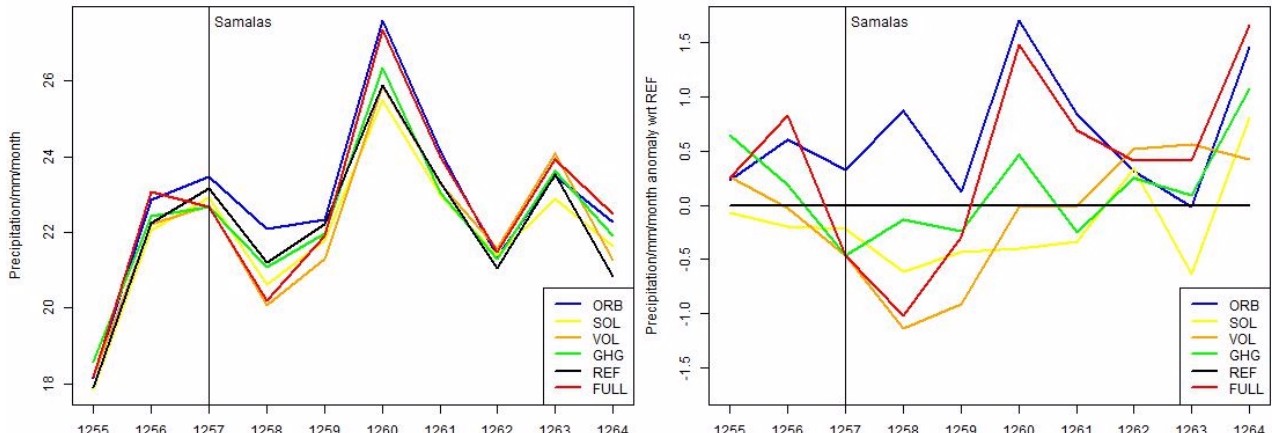

**Figure 11.** Annual mean total monthly precipitation averaged over the EMME domain (left) for the simulations with the different forcings and the same values represented as anomaly (right) with respect to the reference CCLM from 1255 to 1264.

The impact of the different forcings on the seasonal mean precipitation compared to the REF simulation is depicted in Figure 12. While most areas do not present statistically significant differences, according to the Student's t-test ($\alpha$=0.05), most pronounced differences are observed where the total precipitation sum and standard deviation are highest (Figure 10). Across all simulations, there is an increase in precipitation during autumn, with the largest differences observed in the FULL and ORB, especially in the southernmost parts of the domain and in Persia. These differences are related to changes in temperature due to the increased solar radiation (see Figure 4). A similar pattern is observed in summer, albeit with more substantial differences in the southern regions. In spring, negative anomalies are small and only statistically significant here and near the Caspian Sea in the FULL simulation. In contrast, differences in winter are negligible.

In summary, the annual and seasonal distribution of precipitation appears to be less influenced by external forcing compared to the 2m temperature. This is connected to the intricate processes involved in precipitation generation, especially on local to regional spatial scales. It is likely that an ensemble of different global ESM simulations could identify varying years with the lowest precipitation during the simulated decade, regardless of changes in external forcing. Following the volcanic eruption, a slight reduction in precipitation is observed, with more pronounced impacts in simulations that include volcanic forcing (VOL, FULL). Seasonality is also only minimally affected, with a slight increase in precipitation in the southern part of the domain during summer and fall, a trend that is somewhat more pronounced in simulations with orbital forcing (ORB, FULL).

### 3.2.3 Monthly Totals - Distribution and Extremes

In this section, we explore the distribution of total monthly precipitation values for different simulations in each grid box and month in the EMME domain over the decade encompassing the Samalas eruption (Figure 13). The lower quartile of total monthly precipitation is zero in all simulations, reflecting locations with no precipitation throughout the year. The differences in the median range from 0.9 mm/month (SOL) to 1.05 mm/month (REF), while the upper quartile values span from



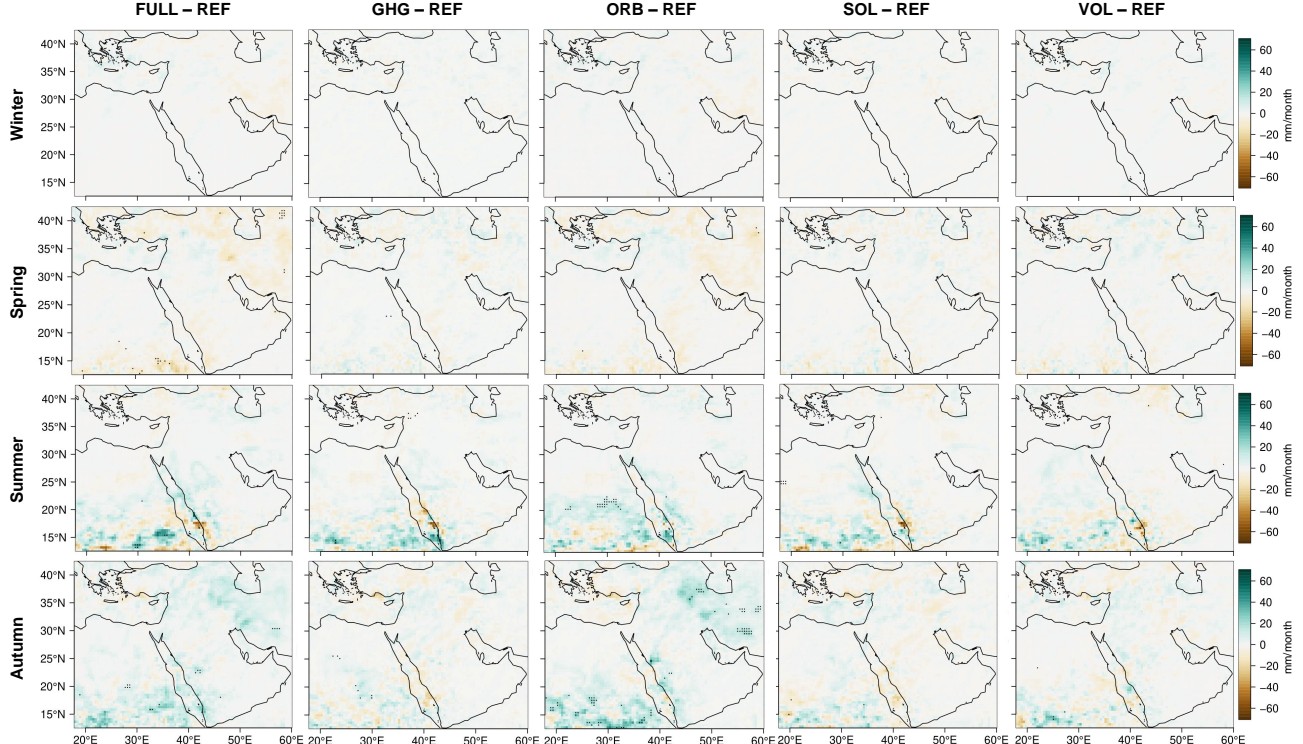

**Figure 12.** Seasonal mean of monthly precipitation sums for the decade 1255-1264 for the different simulations compared to REF. Significant differences are marked with a dot (Student's t-test, $\alpha$=0.05). From left to right - FULL, GHG, ORB, SOL, VOL and from top to bottom - winter, spring, summer, autumn.

21.37 mm/month (SOL) to 22.56 mm/month (ORB), showing minor variations. Outliers vary strongly, with maximum values ranging between 1200 and 1500 mm. The FULL simulation records the highest value, while the ORB and GHG simulations demonstrate a high density of very high values.

Density plots depicting the monthly total precipitation in the REF (red) and FULL (turquoise) CCLM simulations are shown in the center and right parts of Figure 13. However, due to the broad range of monthly precipitation, distinguishing the difference becomes challenging in the plot covering the entire spectrum (center part of Figure 13). Therefore, the x-axis values are constrained to 10 mm in the right part of Figure 13 to facilitate comparison. Given the similar distributions in both simulations, the seasonal scale (again with an x-axis limited to 10 mm) is presented in Figure 14. In summer, the VOL and SOL simulations

exhibit a higher density of zero and low monthly precipitation totals compared to the REF and GHG simulations, while the ORB and FULL simulations demonstrate the lowest density. Equally, in the other seasons, the distributions remain similar across all simulations. In general, spring shows the highest density of zero precipitation, followed by summer, fall and winter. A moderate monthly precipitation total of 2.5 mm displays the highest density in summer, followed by fall and winter, with





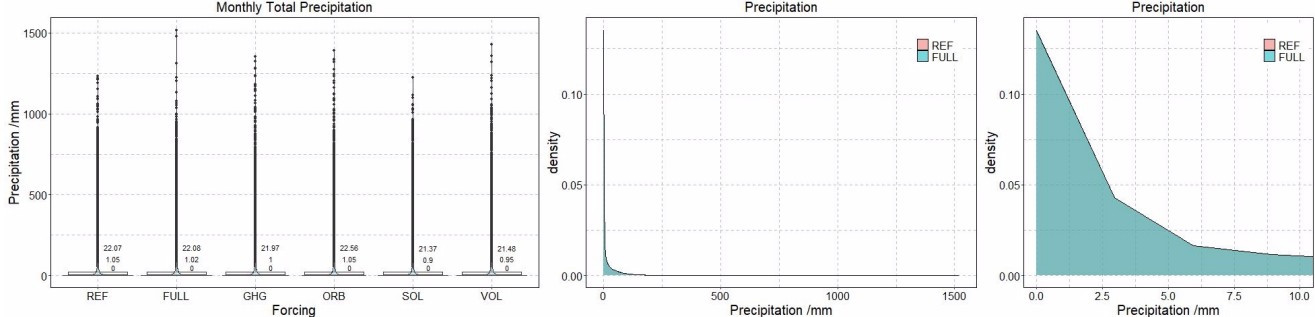

**Figure 13.** Violin plot combined with boxplot (left, values are upper quartile, median, lower quartile) of the precipitation for the different sensitivity experiments and density plot (right) of the 2m temperature for the REF and the FULL CCLM simulation. The right plot is the same as the center but showing a limited x value range.

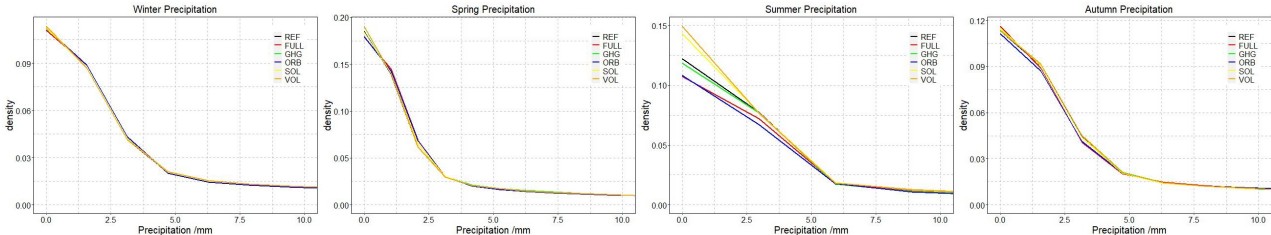

**Figure 14.** Density plot of the monthly precipitation (only shown up to 10 mm/month) for the different forcings in the CCLM and for every season.

the lowest density observed in spring. A precipitation sum of 5 mm is most frequent in summer, with small differences noted
between the other seasons.

The spatial distribution of minimum and maximum values for all simulations compared to REF is shown in Figure C2 in Appendix C. Given that minimum precipitation is expected to be zero across most regions and in all simulations, the differences compared to the REF are consequently very small. Similarly, the differences in the maximum monthly precipitation totals, reaching up to 1500 mm, are also much greater, which is particularly evident in summer and fall, especially in the subtropical
and tropical regions. However, no clear anomaly signal is evident, indicating that these differences are not attributable to physical processes triggered by changes in external forcings.

In addition to the minimum and maximum values of the monthly precipitation totals, Table 3 shows the 5th, 10th, 90th and 95th percentiles. The 90th and 95th percentiles are highest for ORB and FULL simulations, while the highest maximum values are observed in the FULL and VOL simulations. The lower extreme value 0 is inherent to nature and thus influences the 5th
and 10th percentile values. Generally, the FULL and ORB simulations capture the most extreme precipitation values, while the REF, SOL and GHG simulations reflect more moderate values.

In summary, the FULL simulation produces the largest monthly precipitation sums among the test simulations. However, these





**Table 3.** Minimum and maximum total monthly precipitation values across the full area and the whole decade of the different experiments associated with 5th, 10th, 90th and 95th percentiles. The experiment with the most extreme value is in bold.

| Precipitation | | | | | | |
|---|---|---|---|---|---|---|
| Name | Min | 5% | 10% | 90% | 95% | Max |
| REF | 0 | 0 | 0 | 68.43 | 108.60 | 1233.10 |
| FULL | 0 | 0 | 0 | 69.28 | 111.00 | **1517.84** |
| GHG | 0 | 0 | 0 | 68.77 | 109.66 | 1352.45 |
| ORB | 0 | 0 | 0 | **70.17** | **112.30** | 1390.77 |
| SOL | 0 | 0 | 0 | 67.74 | 108.56 | 1226.04 |
| VOL | 0 | 0 | 0 | 68.33 | 108.65 | 1426.89 |

are considered outliers, and thus, the statement cannot be generalized. Nevertheless, since the 90th and 95th percentiles are also highest in the simulations with orbital forcing (ORB, FULL), it suggests that overall the monthly precipitation totals can reach

higher values. When looking at the seasonal precipitation totals, the differences between the simulations are most pronounced in summer.

## 4 Conclusions and Outlook

In this study, we implemented external climate forcings into the regional climate model COSMO-CLM to assess their individual impacts on regional climate, focusing on a substantial volcanic event, the Samalas eruption in 1257. We conducted simulations

spanning ten years using various configurations: the original CCLM setup (REF), each forcing implemented separately, which are greenhouse gases (GHG), orbital (ORB), solar (SOL), and volcanic (VOL) forcing, and a fully forced model (FULL). While solar and greenhouse gas forcings played minor roles due to their limited variability within the chosen period, volcanic forcing had a significant impact, particularly in response to the Samalas eruption.

Direct implementation of the volcanic forcing led to a pronounced cooling effect in the year following the eruption. The orbital

forcing influenced the timing and intensity of the seasons by changing the position of the Earth in relation to the sun on longer time scales in the order of centuries. During the Samalas period, there were slight changes in seasonality compared to the present-day configuration in REF. Our findings indicate that reduced insolation led to lower temperatures in winter and spring, while increased insolation caused a rise in temperatures during autumn. Land-use changes and in particular the choice of external data set have a substantial impact on simulation outcomes. Although not detailed in this study, this was tested,

emphasizing the significance of the land-use dataset. For consistency, we used the original external data created with EXTPAR in this analysis, but for future simulations, we intend to incorporate land-use information from the driving MPI-ESM-LR.

By incorporating external forcings, including solar, orbital, volcanic, greenhouse gas, and land-use changes, we facilitate a more detailed and high-resolution analysis of past regional climatic changes while ensuring that these critical factors are fully accounted for. The simulated 2m temperature strongly connects with elevation, while precipitation patterns are influenced by



topography and coastlines. Both are better represented in the RCM compared to the ESM, resulting in cooler mean temperatures and higher total precipitation amounts in respective regions.

The implementation of external climate forcings presented here will serve as the foundation for an unprecedented 2500-year transient RCM simulation focusing on the Eastern Mediterranean / Middle East region. Once completed, this extensive transient simulation will be subject to further analysis in future studies with various regional, temporal and thematic focuses.

Through the comparison of model outputs with paleo and proxy reconstructions, we will enhance our understanding of the impacts of individual and combined forcings on regional climate, extreme events, and the underlying processes and dynamics. Moreover, the unique new data set, generated through the implementations presented in this study, in conjunction with proxy reconstructions and historical sources for the area, will facilitate a more comprehensive understanding of potential climate-society interactions and study potential causation. This interdisciplinary approach will shed light on how past climate variations

may have influenced societal dynamics, adaptation strategies, and vulnerability to environmental changes.

The methodology presented in this work can be adapted to any period of the past for which external forcing reconstructions are available. However, it is important to note that the significance of different forcings may vary widely depending on the chosen period. This flexibility enables researchers to explore climatic changes and their impacts across different historical contexts, contributing to a more nuanced understanding of the Earth's past climate dynamics.

**Appendix A: Land-Use Change Forcing**

The land surface and land-use is set to a constant map in present-day studies. To implement a transient change of the land-use, the MPI-ESM-LR output is used to produce one of these maps for each simulated year. The variables needed for the land-use in CCLM are shown in Table A1. The plant coverage (PLCOV) and the leaf area index (LAI) are directly calculated with the help of the MPI-ESM-LR output variables $var31$ (PLCOV) and $var107$ (LAI). The variable for PLCOV contains ten

different land cover types (tropical broadleaved evergreen and decidiuous forest, temperate/boreal evergreen and deciduous forest, raingreen and cold shrubs, C3 and C4 perennial grass, crops and pasture). The variable $FOR\_D$ and $FOR\_E$ needed for CCLM are calculated with the forest types, meaning that $FOR\_D$ is the sum of tropical broadleaved and temperate/boreal deciduous forest and $FOR\_E$ is the sum of tropical broadleaved and temperate/boreal evergreen forest. The six land-use variables are used to create an external data file for each year which is then used by the CCLM to have the information about

the land-coverage. Compared to the other forcings, the land-use data appears to cause a larger order of difference on the climate variables. This is because the land-surface information of the external data generated by EXTPAR and the data based on the output of the transient ESM simulation are very different. In the transient simulation the information comes periodically (e.g. yearly) from the land-model JSBACH of the driving model MPI-ESM-LR. To be used by the CCLM the JSBACH output needs to be converted to a specific format, for example by correcting the horizontal resolution. This is done for the variables $LAI$,

$PLCOV$, $FOR\_E$ and $FOR\_D$ where e.g. the plant coverage is a general variable for how much of the ground is covered by plants. This is shown in Figure A1 for the JSBACH output (topleft), the converted climatology (bottomleft) and the EXTPAR data (bottomright). The JSBACH data is divided into different land cover types. Figure A1 shows in the topright the dominant



**Table A1.** Land-use variables in external data created for CCLM.

| variable | meaning |
|----------|---------|
| $PLCOV\_MX$ | Maximum plant coverage |
| $PLCOV\_MN$ | Minimum plant coverage |
| $LAI\_MX$ | Maximum leaf area index |
| $LAI\_MN$ | Minimum leaf area index |
| $FOR\_D$ | Decidiuous forest |
| $FOR\_E$ | Evergreen forest |

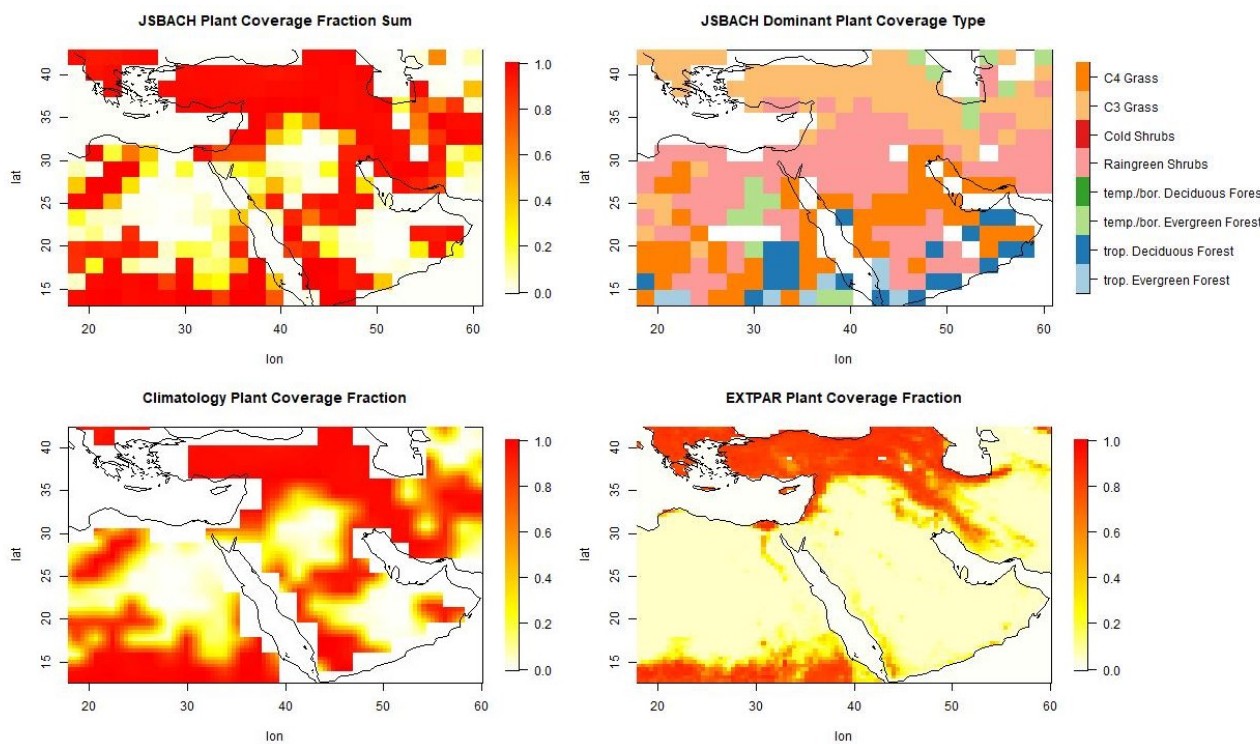

**Figure A1.** Distribution of the plant coverage in the JSBACH output (topleft) for the 1250s as interpolated for the CCLM input (bottomleft) and as given in the EXTPAR external data created for the present. The dominant land cover type for each grid cell in the JSBACH output is in the topright.

land cover type for each grid cell. In most grid cells there are also other land cover types existent but not shown here. The topleft of Figure A1 shows the sum of all those types for each grid cell and the bottomleft shows the same but interpolated to

the EXTPAR grid with a bilinear interpolation to the higher resolved grid.

In the desert regions the differences between the EXTPAR and the Climatology plant coverage are most prominent. Here are mostly growing raingreen shrubs with not very high plant coverage and C3 and C4 grass with higher plant coverage. The





water and $CO_2$ efficient C4 grass leads to a relatively high plant coverage also in regions known as desert. In JSBACH the different land cover types are treated differently also in their effect on the atmosphere. In CCLM plants except for deciduous
and evergreen forest are treated all the same with a seasonal phenological cycle.

To adequately represent the effect of the different forcings without being affected by the external influences, and to warrant a better basis for comparison, we only compare simulations with the same external surface data generated by EXTPAR.

**Appendix B:  Annual distribution maps**

**Appendix C:  Seasonal minimum and maximum distribution maps**





**Figure B1.** 2m temperature yearly distribution of differences between the differently forced CCLM simulations and the reference simulation without explicit forcing. Significant differences are marked with a dot. From left to right - FULL, GHG, ORB, SOL, VOL and from top to bottom - 1255 to 1264.



**Figure B2.** Yearly mean of daily precipitation anomaly for the differently forced CCLM simulations with respect to the reference simulation without explicit forcing. Significant differences are marked with a dot. From left to right - FULL, GHG, ORB, SOL, VOL and from top to bottom - 1255 to 1264.





**Figure C1.** Seasonal minimum (top) and maximum (bottom) 2m temperature distribution for the different simulations with respect to REF. From left to right - FULL, GHG, ORB, SOL, VOL and from top to bottom - winter, spring, summer, autumn.





**Figure C2.** Seasonal minimum (top) and maximum (bottom) (monthly) precipitation distribution for the different simulations with respect to REF. From left to right - FULL, GHG, ORB, SOL, VOL and from top to bottom - winter, spring, summer, autumn.



*Code and data availability.* The COSMO-CLM model is available for all members of the CLM-Community via their website www.clm-community.eu. It is free of charge for all research applications. Either the user needs to be a member of the CLM-Community or the respective institute needs to hold an institutional license. The changes explained here can be directly implemented in the original source code. Detailed model code snippets can be seen in the supplement file. The model version with all forcings here explained implemented can be made accessible for CLM-COmmunity members at the Zenodo repository (10.5281/zenodo.14288621).

The forcing data is published data used for CMIP6/PMIP4 simulations. The greenhouse gas concentrations are published by Meinshausen et al. (2017). The orbital forcing is the yearly dataset by Berger (1978). The solar forcing is published by Jungclaus et al. (2017) and the volcanic forcing by Toohey and Sigl (2017). The simulation results are archived at DKRZ and are available upon request to the authors. The monthly mean temperature and the total monthly precipitation used for this analysis are uploaded to the Zenodo repository (10.5281/zenodo.14397609).

*Author contributions.* EH and MZ developed the model code and performed the RCM simulations. SW performed the ESM simulations. EX and JL supervised the project. EH conducted the formal analysis with the support of EX and SW. EX, SW, MA and EH contributed to the conceptualization of the manuscript. EH prepared the manuscript. All authors contributed with discussion of the results and editing of the manuscript.

*Competing interests.* At least one of the (co-)authors is a member of the editorial board of Climate of the Past. Other than that, the authors declare that they have no conflict of interest.

*Acknowledgements.* This work used resources of the Deutsches Klimarechenzentrum (DKRZ) granted by its Scientific Steering Committee (WLA) under project number bb1201. EX acknowledges support by the Greek "National Research Network on Climate Change and its Impact" (project code 200/937). EX, MA acknowledge support by the German Federal Ministry of Education and Research (BMBF) project NUKLEUS (grant number 01LR2002F).




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
