# Peer review of "On the Implementation of External Forcings in a Regional Climate Model - A Sensitivity Study around the Samalas Volcanic Eruption in the Eastern Mediterranean / Middle East"

_EGUsphere, 2025_

## Author Response (AR1)

**Dear Reviewer #1,**

Thank you very much for very helpful suggestions. We have addressed the comments and suggestions and we have included those in the revised version. Detailed answers to your suggestions can be found below:

*Summary*

*The authors investigate the effect of including the correct external forcing in the regional climate model for past climate states. They found that orbital changes and volcanic forcing has some impact on the temperature in the region of the eastern Mediterranean and the Arabic Peninsula. The precipitation response is rather weak and not significant.*

*General*

*Overall, this is a well structured and well written manuscript, which investigates an interesting technical aspect of paleo climate modelling. The authors showed that the impact of not including the full forcing in the regional climate model (RCM) seems to be of second order relevance compared to the gain of higher resolution although in some variables it might be important to include the forcing in the RCM correctly. Such a clear statement is missing and can be included in the manuscript if the authors agree.*

➢ Thank you very much for the important general suggestion, we insert that statement 'The impact of not including the full forcing in the RCM seems to be of second order relevance compared to the gain of higher resolution although in some variables it might be important to include the forcing in the RCM correctly' into the revised conclusions section.

*Thus, I think this is certainly of importance to the scientific community and I recommend publication after some minor revisions.*

*Major*

1. *Fig. 6: One problem is that the authors only have one simulation each for one forcing. I know that the simulations are expensive, so a small ensemble of say 10 simulations for REF and FULL might not be feasible but at least the authors need to mention this shortcoming in the results past and the conclusions.*

➢ Many thanks for this important major comment on the number of simulations. It is true we have only one simulation for each because we could not have more, due to limited time and costs. We have thus added a brief statement addressing the limitation of using only one simulation per forcing in the discussion of Figures 6 and 11, as well as in the Conclusions. Further, the availability of a single ESM simulation contributed to the limited ensemble size. This fact has also guided our focus on examining the effects of different configurations of external forcings within the RCM setup, rather than assessing the role of internal variability within the combined ESM–RCM framework. This approach is now also explained in a paragraph of the introduction.

*Minor*

*Middle East is maybe problematic as it is a geopolitical concept. Please use a different phrasing for the region. At least some of my colleagues react on using this for a geographical region.*

➢ We selected the naming of our study area as 'Eastern Mediterranean and Middle East' based on existing published work, e.g., Zittis et al. (2022) and the work of

the [EMME-CARE](), a regional Centre of Excellence for climate and atmosphere research in the Eastern Mediterranean and Middle East region of the Cyprus Institute.

*P1,l4: Please change to "Here,"*

*P1,l15: I think that the current study does not provide "a unique source of information for the comparison of paleoclimate simulations with proxy records" as only sensitivity simulations are presented for a short period in the past. The full 2500 year long simulation certainly will do this but for this study I suggest to remove the sentences "The study is part of the new 2500-year-long transient, fully forced RCM simulation over the Eastern Mediterranean / Middle East. This work introduces a unique source of information for the comparison of paleoclimate simulations with proxy records and reconstructions" .*

> ➢ Thank you for this comment. We have reformulated the sentence as:
> "The study is the basis for the new 2500-year-long transient, fully forced RCM simulation over the Eastern Mediterranean / Middle East. It offers an assessment of the implementation of forcings in the RCM, along with an enhancement of the simulations' skill through the use of the RCM."

*P2,l44: For LGM there are also a couple of studies using even convection permitting resolutions:. It may be nice to mention this here.*

*Velasquez P., J. O. Kaplan, P. Ludwig, M. Messmer, and C. C. Raible, 2021: The role of land cover on the climate of glacial Europe. Climate of the Past, 17, 1161-1180.*

*Velasquez, P., Messmer, M., and Raible, C. C., 2022: The role of ice-sheet topography on the Alpine hydro-climate at glacial times, Climate of the Past, 18, 1579–1600.*

*Russo, E., Buzan, J., Lienert, S., Jouvet, G., Velasquez, P., Davis, B., Ludwig, P., Joos, F. and Raible, C. C., 2023: High resolution reconstruction of LGM climate over Europe and the Alpine region using WRF, Climate of the Past, 20, 449–465, https://doi.org/10.5194/cp-20-449-2024 .*

> ➢ Thank you very much for these suggestions. We have introduced them in the manuscript.

*P3,l66: please change "Section 2 details the models, their configurations,"*

*P3,l68: "Section 3 focusses " reads better.*

*P3,l71: "In Section 4,  we present the main conclusions and outline potential directions for future research." Reads better.*

*P3-4: Section 2.1 I would suggest to include part o the discussion presented in this section already in the introduction to make the motivate why the authors focus on this region and this period.*

> ➢ Thank you for the suggestion. We moved large parts of the description of the volcanic eruption and the domain to the introduction. The motivation to focus this area and period is moved to the introduction while the more detailed background information stays in the Material and Methods section.

*P5,l120: Please remove "is used" after (Tiedtke, 1988).*

*P5,l126 "In this study, only the"*

*Caption Fig 3: "Transient forcings ..."*

*Fig.5 For the mean and the STDD plots it would be better to use discrete colours rather than a continuous colour scale.*

➢ We have updated the color palette in Fig. 5 and Fig. 10, thank you for this comment.

*L191-195: This is a rather lengthy description for textbook knowledge so just write "The better representation of lower temperatures over orography is due to the lapse rate effect."*

➢ We have replaced the lengthy description with the suggested sentence.

*Section 3.1.2/ Fig.6: It would be interesting to also include the ESM result so we can see how the RCM changes the ESM response to volcanic eruptions. Maybe this is interesting for all plots.*

➢ ESM results have been added to Fig. 6, as well as Fig. 11. For the other plots, we decided to focus on the pure sensitivity study of the RCM with different forcings implemented.

*Fig 7: I would suggest using a non parametric test as the authors have really a small number of years, So Mann-Whitney-U test would be better here.*

➢ The Mann-Whitney-U test (here called Wilcoxon test according to the used R-function performing the Wilcoxon rank sum test) has now been introduced in Fig. 7 and Fig. 12 instead of the student's t-test. Thank you very much for the hint.

*L315ff: I think the authors suggest that the decrease in precipitation in 1258 is due to volcanic forcing but given the variability of the time series I am not convinced that there is causality. For this statement the authors need to run an ensemble of simulations. So please rephrase.*

➢ Thank you for the comment. We now added a statement about this at the end of the paragraph as follows: "There is no indication of a link between the volcanic or any other forcing and precipitation. The timeseries shows the natural variability of precipitation within a decade. Due to limited resources, ensemble simulations or longer timeseries were not carried out, which would have made it possible to distinguish between natural fluctuations and external forcing."

*Section 3.2.3 I guess it is always mm/month, please adapt this in the manuscript and the figures.*

➢ We have adapted mm/month in figures and manuscript.

*Table 3: It makes no sense to show Min, 5% and 10% as precipitation is not Gaussian distributed.*

➢ Thank you very much for this comment! We now present 50th, 75th, 90th, 95th, 99th percentiles and maximum value instead.

*L389: I would not fully agree that the effect of volcanic forcing in the RCM is pronounced. With just one simulations the authors need to be a bit more careful.*

➢ We have modified the sentence to "Direct implementation of volcanic forcing led to an additional cooling effect after the eruption in the test simulations." And discussed the issue of having only one simulation in various sections of the manuscript including the Conclusions section.

*L393-395: "Land-use changes and in particular the choice of external data set have a substantial impact on simulation outcomes. Although not detailed in this study, this was tested, emphasizing the significance of the land-use dataset". This is fully new result and not presented in the main paper, so I recommend including this result or remove the sentence.*

➢ Thank you for this important comment. We removed the sentence and adjusted the sentence before to: "Land-use changes and in particular the choice of external data set have a substantial impact on simulation outcomes. Because the focus of the manuscript was on the the Samalas volcanic eruption those effects were not discussed in greater detail." Including these results would mean an improper extension of this manuscript to a completely different topic and will be presented in an additional publication.

*411: please change Methodology to Method*

➢ Thank you very much for the above list of comments. We implemented all suggested changes in wordings and structure of sentences into the revised version.

We greatly appreciate your support and hope that the revised version now fully meets your expectations and requirements.

Yours sincerely, on behalf of all co-authors

Eva Hartmann

**Dear Reviewer #2,**

Thank you very much for very helpful suggestions. We have addressed all points and included them in the revised version. Please see below point-by-point responses to the comments:

*This detailed study by Hartmann et al. examines the use of a climate model output (MIP-ESM-LR) to drive a higher-resolution regional climate model (COSMO-CLM), where the external forcing implemented follows Jungclaus et al., 2017. Apparently this study is part of a 2500 year regional climate model simulation over the Eastern Mediterranean and the Middle East region that I agree is an important region in many aspects. The selected time period for this particular study is one decade and covers the 1257 Samalas eruption. The contribution of each forcing in the simulated regional climate variability is tested through several single-run sensitivity simulations (single and combined forcing using the MPI-ESM output) in addition to a present-day COSMO-CLM reference simulation. It is mentioned that this method leads to a novel paleo-regional climate model and this paper is dedicated to distinguishing between the relative contribution of each forcing and the internal variability with the aim to see if the incorporation of external forcing into the regional climate model enhances realism at the regional scale.*

*The methodology used is good in general, biases and detected (regional) climate extremes associated with each forcing are studied in detail, and the results have the potential to contribute to the overall enhancement of reliability within climate and regional model communities on various timescales. This manuscript is also well structured and reads well for the most part and so I would say that it is worthy of being published in Climate of the Past after the authors address some of my concerns. Although I suggest major revision, it is mainly due to lack of discussions regarding certain aspects and/or requested clarification regarding parts of the methodology - but the authors can of course use these suggestions to go as deep as they see fit.*

*These concerns mainly relates to the fact that these single-run sensitivity simulations are only but one realisation of climate variability over the study area and this needs to be addressed in more details and how this might impact the results obtained (and if this will be accounted for in the larger 2500yr run). This becomes quite clear in Figure 6 and 7. In Fig. 7 it seems that the cooling that followed Samalas did not contribute significantly to the mean seasonal temperature anomalies (in any season) in the FULL experiment. As can also be seen in Figure 6, the T anomaly decrease is small, -0.4°K (perhaps within the range of natural climate variability), so if the authors were to run an ensemble of these simulations, result might only show a weak significant T decrease in 1258.*

> ➢ We now explicitly address the fact that the analysis is based on a single realization within the range of natural variability in several paragraphs throughout the results, especially for Fig. 6 and Fig. 11, and in the conclusions section. We had limited time and the implementation of the ESM simulations is also afflicted with high costs. Here we focused on the effect of different configurations of the external forcings in the RCM. Another reason for this is the limitation to just one ESM simulation. We now address this issue in a paragraph of the introduction. Overall, with the sensitivity analysis, which only contains one simulation in each case, we were not able to represent the entire potential bandwidth of simulated climate variability, but we were at least able to recognize a tendency as to what influence the various forcings have.

*Also, the reference period is highly important. Here the reference period (for the RCM) is tuned according to present-day climate, did the authors check how/if the results would change according to pre-industrial reference period? I think that could give us important information on how relevant these post-volcanic anomalies were with respect to the current time period, I urge the authors to at least discuss this. It would also make sense to discuss the response detected here and compare with the known climate impact of Samalas partly discussed in the intro (maybe in discussions). I feel that too much focus is put on absolute temperature changes when referring to other studies with respect to the results.*

> ➢ The focus of our work is on the differences between the external forcing configurations implemented to the RCM rather than explicitly addressing the

influence of the Samalas volcanic eruption or the impact of volcanic eruptions in general on climate and the study area. Focusing on these differences, our results are more or less independent on the reference period selection. We added more information on the reference simulation in the section 2.5 Experiments and the Conclusions, where we no additionally refer to literature from the introduction. The RCM CCLM is originally not configured for climate simulations of pre-industrial periods. This means that for a pre-industrial reference we would have needed already the changes we implemented throughout this study. Therefore, we took the standard CCLM configuration as reference. As discussed, this is particularly evident in the simulations with implemented orbital forcing. The impact would have been even greater if the case study was from further in the past. In addition, we shifted the focus from the absolute values to the variability and now place it in a larger context at some points in the discussion.

*It is also interesting that the seasonal mean of precipitation is more significant (summer/autumn) than the seasonal mean temperature, since precipitation is usually more sensitive to parametrisation in these models and thus less robust compared to simulated temperature. How well in general does the regional model simulate precipitation (compare to other models/obs/etc)?*

➢ We agree that the differences for summer/autumn show larger differences for precipitation than for the other seasons. Still, the differences are mostly not statistically significant at the 0.05 level (also when using the non-parametric Wilcoxon Rank Sum test). The larger differences for specifically those regions in the southern part of our domain could be connected to the variability of the ITCZ, with its most northern position during northern summer. Considering the statistical significance of the differences, only very few regions are marked as statistically significant. For near-surface temperature, the statistically significant areas are however substantially larger in extent, indicating the higher sensitivity of temperature on changes in external forcings compared to precipitation.

*Here below are more specified comments:*

*L53: „Various external forcing become relevant, ...„. This sentence sounds incomplete, become relevant in regional climate variability?*

➢ Thank you very much for the comment. We completed the sentence with: "The relevance of various external climate forcings for climate variability depends on the period under consideration."

*L120: remove „is used".*

➢ We removed it.

*L134: Indeed volcanic forcing can have decadal climate impacts through secondary processes beyond the well-known direct surface cooling.*

➢ Thank you for the important hint. We added a statement here: "In contrast, volcanic forcing has a direct strong, rapid and short-term impact (Wigley et al., 2005) and up to decadal climate effects through secondary processes (Kremser et al., 2016)."

*L142: maybe hundreds of millennia? Eccentricity oscillates in periods of 100.000 to 400.000yr.*

➢ Thank you, we changed to: "Orbital forcing is represented by the eccentricity, the obliquity and the longitude of perihelion, varying on time scales of dozens to hundreds of millennia."

*L143: Annually -> Annual ?*

➢ We changed it.

*L208: 1256 is also more than 1K cooler than in 1255. Although the direct temperature response is interesting, the anomalies reveal more about the significance of this T decrease.*

➢ Thank you very much for the important comment. We added a statement about it at the end of the paragraph: "The post-eruption cooling is physically meaningful, but given the 1 K cooling prior to the eruption (1255 to 1256), it is not significant if only this one decade is considered. This issue will be addressed in future studies involving a multi-century transient simulation. Another way to decipher the impact of external versus internal/natural forcing is to carry out an ensemble of simulations. However, this was not possible due to restricted computational resources. Here, we focus on the differences between the implemented forcings within the RCM for single simulations."

*L220: should this not be 1262 as in Fig. 6?*

➢ Yes, it should. We changed it.

*L293: So in general, first the well-known precipitation decrease followed by a modified/reversed pattern? Could you reflect on the timescale these changes might act on? This is also related to the comment above in L134.*

➢ Thank you for the reminder. We also added a statement here: "On the regional scale, this pattern might be modified or even reversed because of changes in major regional circulation systems (i.e. monsoon, changes in ITCZ), whose influence on the regional climate is considerably larger. It is therefore possible that the influence of the volcanic eruption on precipitation variability will only become visible years later through secondary processes and feedback mechanisms."

*L323-324: This could depend on the potential modified patterns followed by the strong surface cooling, so maybe it does not disappear but it certainly become less dominating.*

➢ Thank you for the comment. We adjusted the sentence as follows: "Due to the increased complexity of potentially modified general circulation patterns caused by the surface cooling, the influence of the volcanic eruption on precipitation variability becomes less dominant after one to two years."

*Figure 1 caption: Does this mean that this is the author's representation of Toohey and Sigl?*

➢ Yes it does. We changed the sentence to: "Own representation of data from Toohey and Sigl (2017)."

We greatly appreciate your support and hope that the revised version now fully meets your expectations and requirements.

Yours sincerely, on behalf of all co-authors

Eva Hartmann

---

## Author Response (AR2)

**Dear Prof. Goosse,**

Thank you very much for accepting our manuscript for publication. We have addressed the comments and suggestions made by the reviewer and we have included those in the revised version. Detailed answers to the comments can be found in the second part of this document.

We remain very grateful for your support and hope that our manuscript now meets all the requirements for publication.

Yours sincerely and with best regards, on behalf of all co-authors

Eva Hartmann

**Dear Reviewer #2,**

Thank you very much for the positive feedback! We have gladly incorporated the further suggestions for improvement into our manuscript. Details follow below:

I'm happy to see that this manuscript has been substantially improved. I only have a few minor comments listed here below and after they have been addressed I recommend it to be publishable in Climate of the Past.

L76: Not sure outbreaks is correct here. Maybe active volcanic periods or something similar.
- ➤ You are right. We have changed the sentence to: "Periods of high volcanic activity are prominent candidates for conducting sensitivity experiments over the last few millennia." Additionally, we adjusted the "outbreak" in one of the next sentences to: "The effects of the Samalas eruption were felt throughout the Mediterranean and contributed to considerable cooling and existential crises in various regions."

L167: A lot of citations missing here. Either, a citation to an overview paper like Marshall et al (??) (and write "see reference therein" etc.) or cite more papers would make sense. Example: Stenchikov et al., 2009; Shindell et al., 2009; Otterå et al., 2010; Zanchettin et al., 2012; Swingedouw et al., 2015
- ➤ Yes, that's correct. We have included all the references you mentioned here.

L294: rephrase to „...which is within the range of natural variability" or similar.
- ➤ Thank you very much for the suggestion. We modified the sentence to: "In contrast, the SOL, VOL and GHG simulations show only minor, non-statistically significant temperature anomalies, ranging from slightly negative to positive values below 0.5 K, which is within the range of natural variability."

L336-337: confusing sentence, please rephrase („results in theory in...")
- ➤ Yes, indeed this sounds confusing. We changed the sentence to: "After a major volcanic eruption, this theoretically leads to lower atmospheric humidity and reduced precipitation on a global scale."

We greatly appreciate your support and hope that the final version fully meets your expectations and requirements.

Yours sincerely, on behalf of all co-authors

Eva Hartmann